# Impact of Maturation on Myocardial Response to Ischemia and the Effectiveness of Remote Preconditioning in Male Rats

**DOI:** 10.3390/ijms222011009

**Published:** 2021-10-12

**Authors:** Lucia Kindernay, Veronika Farkasova, Jan Neckar, Jaroslav Hrdlicka, Kirsti Ytrehus, Tanya Ravingerova

**Affiliations:** 1Institute for Heart Research, Centre of Experimental Medicine, Slovak Academy of Sciences, 9 Dúbravská cesta, 84104 Bratislava, Slovakia; lucia.griecsova@gmail.com (L.K.); weroro@gmail.com (V.F.); 2Institute of Physiology, Academy of Sciences of the Czech Republic, 1083 Vídeňská, 142 00 Prague, Czech Republic; Jan.Neckar@fgu.cas.cz (J.N.); Jaroslav.Hrdlicka@fgu.cas.cz (J.H.); 3Department of Medical Biology, UiT The Arctic University of Norway, 18 Hansine Hansens veg, 9019 Tromsø, Norway; kirsti.ytrehus@uit.no

**Keywords:** ischemia/reperfusion injury, remote ischemic preconditioning, maturation, protective cell signaling

## Abstract

Aging attenuates cardiac tolerance to ischemia/reperfusion (I/R) associated with defects in protective cell signaling, however, the onset of this phenotype has not been completely investigated. This study aimed to compare changes in response to I/R and the effects of remote ischemic preconditioning (RIPC) in the hearts of younger adult (3 months) and mature adult (6 months) male Wistar rats, with changes in selected proteins of protective signaling. Langendorff-perfused hearts were exposed to 30 min I/120 min R without or with prior three cycles of RIPC (pressure cuff inflation/deflation on the hind limb). Infarct size (IS), incidence of ventricular arrhythmias and recovery of contractile function (LVDP) served as the end points. In both age groups, left ventricular tissue samples were collected prior to ischemia (baseline) and after I/R, in non-RIPC controls and in RIPC groups to detect selected pro-survival proteins (Western blot). Maturation did not affect post-ischemic recovery of heart function (Left Ventricular Developed Pressure, LVDP), however, it increased IS and arrhythmogenesis accompanied by decreased levels and activity of several pro-survival proteins and by higher levels of pro-apoptotic proteins in the hearts of elder animals. RIPC reduced the occurrence of reperfusion-induced ventricular arrhythmias, IS and contractile dysfunction in younger animals, and this was preserved in the mature adults. RIPC did not increase phosphorylated protein kinase B (p-Akt)/total Akt ratio, endothelial nitric oxide synthase (eNOS) and protein kinase Cε (PKCε) prior to ischemia but only after I/R, while phosphorylated glycogen synthase kinase-3β (GSK3β) was increased (inactivated) before and after ischemia in both age groups coupled with decreased levels of pro-apoptotic markers. We assume that resistance of rat heart to I/R injury starts to already decline during maturation, and that RIPC may represent a clinically relevant cardioprotective intervention in the elder population.

## 1. Introduction

Ischemic heart disease (IHD) is the main cause of death worldwide, and the global prevalence of IHD is expecting to rise [1]. The research has been, therefore, focused on various methods of cardiac protection and on the clarification of its molecular basis and signaling mechanisms in relationship with aging. Ischemic preconditioning (IPC) is a short-term phenomenon protecting myocardium of all animal species, including humans, against sustained ischemia based on the adaptation to some moderate stressful stimuli [2]. Despite the robust effect of IPC, its application in clinical conditions has limitations since it requires an invasive intervention to get access to the coronary arteries and has a short-term duration. An alternative and more affordable cardioprotective strategy is to apply a stimulus to an organ or tissue that is remote from the heart. This phenomenon has been defined as remote IPC (RIPC), and its cardioprotective effect has been already demonstrated in clinical settings, such as operation of congenital heart defects or coronary artery bypass grafting surgery (CABG) [3,4,5], not only in adults but also in elderly patients [4,6,7], albeit with different results. The constantly growing average life expectancy may lead to an increased occurrence of comorbidities, such as hypertension or diabetes mellitus, that aggravate ischemic injury [8]. In addition, age itself as an independent risk factor that may cause molecular, structural or biochemical changes in the cardiovascular system [9], and powerful cardioprotective strategies effective in young animals lose their effectivity with aging [10]. Therefore, the search for new therapeutic strategies is urgently needed [11]. However, age-dependency of the RIPC has not been explored in detail in experimental settings.

Cardioprotection conferred by different forms of “conditioning” is dependent on the activation of “pro-survival” protein kinase cascades, including RISK (reperfusion induced salvage kinase) cascade involving kinases such as phosphoinositide 3-kinase (PI3K)/protein kinase B (Akt), extracellular signal-regulated kinases1/2 (ERK1/2), protein kinase Cε (PKCε), endothelial nitric oxide synthase (eNOS) [12] and the inactivation of glycogen synthase kinase-3β (GSK3β) [13]. The latter is considered as a crucial step related to the inhibition of mitochondrial permeability transition pore (MPTP) opening, leading to cell death/survival.

IPC involves a “trigger phase” in which proteins are activated during and shortly after IPC induction, and a “mediator phase” in which some of the same proteins are activated during post-ischemic reperfusion [14]. In our previous study, we have demonstrated that attenuation of cardioprotective effects of classical IPC in the hearts of older animals compared with the IPC effects in the younger ones was associated with reduced Akt phosphorylation, decreased expression of PKCε and eNOS as well as with a failure of IPC to upregulate these proteins both at baseline and post-IR [15]. However, it has not been verified whether such a response is also manifested during RIPC intervention, in particular, in relationship with age.

It is believed that the mechanisms of RIPC in the heart are similar to those behind the cardioprotective effect of classical IPC [16], and age-related effectiveness of RIPC has been also demonstrated in some studies [17,18]. However, only few investigators have explored the changes in the efficiency of IPC in the relatively early period of the life span in the process of maturation [15,19,20]. On the other hand, it is known that only a few months of rat’s life represent several years of human life [21]. We assumed that even in the seemingly young rats, significant changes in the resistance of their hearts to I/R injury and also in their response to RIPC may occur during the process of maturation.

Therefore, the present study aimed: (i) to elucidate the effect of age (maturation) on the effectiveness of remote ischemic preconditioning against I/R injury in male rat hearts; (ii) to identify signaling pathways involved in cardioprotection by RIPC in association with age; and (iii) to clarify the effect of RIPC through examination of the levels of pro-survival proteins under baseline (pre-ischemic) and post-ischemic conditions.

## 2. Results

### 2.1. Effect of Aging on the Basic Biometric Parameters of Rats

Aging markedly influenced biometric parameters of rats. Both body weight and heart weight were significantly elevated during maturation with the highest increase observed in the period between the third and sixth month of age (Table 1).

### 2.2. Characteristics of Isolated Hearts: Effect of Age/Maturation on Functional Parameters of Isolated Rat Hearts

The values of hemodynamic parameters in the 3- and 6-month-old groups are summarized in Table 2. There were no significant changes in the values of these parameters with age.

### 2.3. The Effect of Age and Remote Ischemic Preconditioning on Ischemia/Reperfusion Injury

Increasing age markedly influenced the extent of lethal ischemic injury (size of infarction) after ischemia/reperfusion in the control (non-RIPC) group. At the age of 6 months, infarct size significantly increased in comparison with that in the 3-month-old group (6 mo: 33.7 ± 1.8 vs. 28.9 ± 1.0%; # *p* < 0.05, vs. 3 mo; Figure 1A). In the group of animals subjected to RIPC, the size of infarction was also greater in the hearts of the elder group in comparison to those in the younger group (6 mo: 22.9 ± 3.3 vs. 11.7 ± 3.2%, #—*p* < 0.05 vs. 3 mo; Figure 1A). On the other hand, RIPC significantly reduced IS not only in the hearts of the younger group but also in the group of elder animals as compared with respective non-RIPC controls, suggesting preservation of the IS-limiting effect of RIPC (Figure 1A).

No age-related changes in the post-ischemic recovery of LVDP were observed neither in the non-RIPC controls nor in the RIPC-treated groups (Figure 1B). On the other hand, RIPC markedly improved the recovery of LVDP in both age groups of animals. Significantly better recovery of LVDP was observed in RIPC groups compared with the respective control groups (76.6 ± 4.3% vs. 53.5 ± 5.6% at 3 mo; 70.8 ± 8.5% vs. 49.7 ± 5.2% at 6 mo; * *p* < 0.05, RIPC vs. C).

Furthermore, age-related changes in the occurrence of reperfusion-induced ventricular arrhythmias were observed in the hearts of control animals (Figure 1C). In the control group of 6-month-old animals, a significant increase in the total duration of ventricular tachycardia (VT) was recorded, in comparison with that in the 3-month-old control group. Antiarrhythmic effect of RIPC was manifested by a significant reduction in the total duration of VT, when comparing RIPC groups with the controls at 3 months (VT: 38.4 ± 9.2 vs. 84.0 ± 17.0 s; * *p* < 0.05, RIPC vs. C) and also the 6-months-old controls (VT: 71.6 ± 42.3 vs. 204 ± 73 s; * *p* < 0.05, RIPC vs. C) (Figure 1C).

### 2.4. The Effect of Age on the Changes in the Cell Signaling of Remote Ischemic Preconditioning

#### 2.4.1. The Effect of Age and Remote Ischemic Preconditioning on the Expression of Selected RISK Pathway Proteins in Rat Myocardium at Baseline Conditions

Aging affected all studied proteins in the rat myocardium at baseline (pre-ischemic) conditions. In the 6-month-old control group, there was a significant decrease in the phosphorylation (activation) of Akt (Figure 2A) expressed as a ratio of p-Akt and Akt, and in phosphorylation (inactivation) of GSK3β, expressed as a ratio of p-GSK3β and GSK3β (Figure 2C). The expression of eNOS (Figure 3A) and PKCε (Figure 3C) proteins in the 6-month-old group was also decreased in comparison with 3-month-old controls. In the hearts of RIPC-treated animals, only p-GSK3β/total GSK3β ratio was decreased in the elder group (by 25%, Figure 2C), while PKCε levels were increased by 18% (Figure 3C).

In the hearts of younger animals, RIPC did not change p-Akt/total Akt ratio (Figure 2A) and eNOS protein levels (3A), but decreased PKCε levels in the cytosolic fraction (by 24%, Figure 3C). At the age of 6 months, no significant changes in p-Akt/Akt ratio (Figure 2A), eNOS (Figure 3A) and PKCε (Figure 3C) were observed in the RIPC groups compared to non-RIPC groups of the same age. However, phosphorylation of GSK3β was significantly increased by 8% (Figure 2C) in the RIPC group of younger animals, as compared with that in the non-RIPC controls at baseline conditions. Similarly, p-GSK3β/GSK3β ratio was also increased (by 19%, Figure 2C) in the elder RIPC group as compared with the 6-month-old control one at baseline.

#### 2.4.2. The Effect of Age and Remote Ischemic Preconditioning on the Expression of Selected RISK Pathway Proteins in Rat Myocardium after Ischemia and Reperfusion

All studied proteins were also negatively affected by age after ischemia and reperfusion. In the 6-month-old control group, p-Akt/Akt ratio was reduced by 22% (Figure 2B), p-GSK3β/GSK3β ratio by 6% (Figure 2D), eNOS protein levels by 28% (Figure 3B) and PKCε levels by 33% (Figure 3D) in comparison with those values in the 3-month-old group.

In the hearts of the 6-month-old RIPC animals compared to the 3-month-old ones, a significant decrease in the ratio of p-Akt/Akt (by 23%; Figure 2B), p-GSK3β/GSK3β (by 23%; Figure 2D) and in the levels of PKCε (by 41%; Figure 3D) was observed, while the levels of eNOS (Figure 3B) were not changed.

RIPC significantly increased p-Akt/Akt ratio (by 21%; Figure 2B), p-GSK3β/GSK3β ratio (by 41%; Figure 2D), expression of eNOS (by 54%; Figure 3B) and PKCε (by 32%; Figure 3D) in the hearts of younger animals, as compared with respective non-RIPC controls. Importantly, in the elder animals, RIPC also significantly increased p-Akt/Akt ratio (by 19%; Figure 2B), expression of eNOS (by 146%; Figure 3B) and PKCε (by 18%; Figure 3D). Moreover, in the 6-month-old group, there was also a marked trend of enhanced p-GSK3β/GSK3β ratio (Figure 2D) compared with non-RIPC controls.

### 2.5. The Effect of Age and Remote Ischemic Preconditioning on the Expression of Selected Proteins of Pro- and Anti-Apoptotic Cascades in Rat Myocardium

#### 2.5.1. The Effect of Age and Remote Ischemic Preconditioning on the Expression of Selected Proteins of Pro- and Anti-Apoptotic Cascades in Rat Myocardium at the Baseline Condition

Investigation of the expression of pro-apoptotic pro-caspase 3, caspase 3 and the ratio of pro-apoptotic BAX and anti-apoptotic Bcl-2 (BAX/Bcl-2) at the baseline (pre-ischemic) condition showed that in the control elder group, there was a significant increase in pro-caspase 3 (by 8%; Figure 4A), caspase 3 (by 14%; Figure 4C) and BAX/Bcl-2 ratio (by 26%; Figure 5A) compared to the 3-month-old group. In the 6-month-old RIPC group compared to the younger one, significant changes were observed only in the pro-caspase 3 levels (Figure 4A).

RIPC significantly decreased protein levels of pro-caspase 3 (by 12%; Figure 4A), caspase 3 (by 16%; Figure 4C) and BAX/Bcl-2 ratio (by 11%; Figure 5A) in the hearts of younger animals. At 6 months of age, RIPC significantly decreased the levels of caspase 3 (by 22%; Figure 4C) and BAX/Bcl-2 ratio (by 38%; Figure 5A), in comparison with their levels in the respective non-RIPC controls.

#### 2.5.2. The Effect of Age and Remote Ischemic Preconditioning on the Expression of Selected Proteins of Pro- and Anti-Apoptotic Cascades in Rat Myocardium after Ischemia and Reperfusion

In control animals, no impact of age on the expression of pro-caspase 3 (Figure 4B) and caspase 3 (Figure 4D) was observed after ischemia and reperfusion, while BAX/Bcl-2 ratio was significantly increased by 43% (Figure 5B). In the elder group of RIPC-treated animals, only pro-caspase 3 levels were significantly lower (by 19%) as compared to its levels in the younger group (Figure 4B).

RIPC significantly decreased the levels of caspase 3 (by 19%; Figure 4D) and BAX/Bcl-2 (by 8%; Figure 5B) in the younger animals, as compared with non-RIPC controls. In the elder group, RIPC induced a significant decrease in the levels of all proteins: pro-caspase 3 (reduced by 18%; Figure 4B), caspase 3 (reduced by 16%; Figure 4D) and BAX/Bcl-2 ratio (reduced by 40%; Figure 5B) as compared with the non-RIPC controls.

## 3. Discussion

The impact of age as an independent risk factor for cardiovascular diseases has been demonstrated at all levels, from the integral organism, the whole organ/tissue to the changes in cardiomyocytes and DNA [22]. Therefore, we first examined whether basic biometric values and hemodynamic parameters of the isolated rat heart were modified during maturation (Table 1 and Table 2). Although body weight and heart weight were significantly increased in 6-month-olds compared to 3-month-olds rats, no significant changes in the values of hemodynamic parameters were observed, which is in accordance with the studies that did not show age-dependent changes in basic hemodynamic parameters [15,23,24].

In agreement with these studies [15,23,24], we have demonstrated that increasing age caused an increase in the size of infarction in the controls and also in the RIPC-treated animals (Figure 1A). On the other hand, an IS-limiting effect of RIPC was manifested in both age groups. Our results are in agreement with recent studies demonstrating the IS-sparing effect of RIPC in younger (3-month-old) animals [25,26,27,28,29,30,31,32,33]. Only few studies demonstrated the impact of age on RIPC efficiency. In the study by Behmenburg et al. [18], the IS-limiting effect of RIPC in younger rats (2–4 months old) was lost at the age of 22–24 months. However, their older group of animals was much older than the animals used in our study. Furthermore, in a recent study by Heinen et al. [34], the authors confirmed a positive effect of RIPC (human-to-rat RIPC plasma transfer) only in younger rats (2–3 months old) but not in the old ones (22–23 months old).

We did not find any age-dependent changes in the post-I/R recovery of LVDP, neither in the control nor in the RIPC groups (Figure 1B), and confirmed previous findings showing no changes in the post-I/R recovery of LVDP with age in controls [15,35]. On the other hand, some studies demonstrated an age-related reduction in LVDP recovery [23,24]. However, in these studies, the effect of RIPC was studied in animals with a larger age range (4–5 vs. 24 months). In the present study, RIPC was protective in both age groups of rats (3- and 6-month-olds) as compared to their respective controls (Figure 2B). These findings are in accordance with several studies that have demonstrated the protective effect of RIPC on LVDP recovery in younger (3-month-old) rats [26,27,31]. Based on the results of the present study, we might suggest that during maturation, RIPC efficiency was preserved.

With increasing age, a higher incidence of tachyarrhythmias was encountered. This is in concert with a number of studies that demonstrated an elevated susceptibility of the heart to reperfusion arrhythmias with increasing age [20,36,37]. On the other hand, we found that RIPC resulted in a decrease in the total duration of ventricular tachycardia in both younger and elder groups, as compared to the non-RIPC controls (Figure 1C), i.e., maturation did not abolish the antiarrhythmic effect of RIPC. Our data are consistent with other studies that have also demonstrated an antiarrhythmic effect of various forms of RIPC in younger animals [30,38,39], and suggest that maturation did not attenuate the protective effect of RIPC that was maintained in the elder group of rats. However, the mechanisms of this effect require further elucidation.

### 3.1. Cell Signaling of RIPC in Relation with Maturation

The last goal of our study was to characterize the effect of RIPC on the expression of selected cell survival proteins and proteins involved in pro- and anti-apoptotic cascades in relation to maturation, and to clarify whether RIPC shows a “biphasic” effect.

We found that maturation significantly affected all studied cell survival proteins (Figure 2A,C and Figure 3A,C) in the myocardium of control rats already under baseline pre-ischemic conditions, however, in a different way. Our results are consistent with several studies demonstrating the effect of age on proteins related to cell survival. The study by Iemitsu et al. [40] has shown a significant reduction in the levels of p-Akt and p-eNOS with age (4- vs. 23-month-old rats). A decrease in p-Akt and p-eNOS was also found in mice (4 vs. 12 months) [41]. In addition, a reduction in PKCε levels has been demonstrated in experiments comparing neonatal and adult rat hearts [42], or 6- and 22-month-old rat hearts [43]. On the other hand, a study by Whittington et al. [35] reported an increase in Akt phosphorylation under baseline conditions at the age of 18 months, as compared with that in 3-month-old animals, whereas Kostyak et al. [23] and Zhu et al. [44] found that p-GSK3β levels did not change with age (4- vs. 24-month-old rats). A possible explanation for the differences in these results may be related to the rat strains (Fisher 344 vs. Wistar) or to a larger age difference between the groups of animals or tissue sampling techniques that could reflect different subcellular pools of the proteins. Thus, it is still not clear whether and when the changes in the degree of phosphorylation of Akt and GSK3β occur between the 4th and 24th month of age, which may also depend on factors like receptor density and metabolic status.

Both in the group of younger animals, as well as in the 6-month-old group, efficacy of RIPC under baseline conditions was not sufficient to increase Akt activation (Figure 2A), expression of eNOS (Figure 3A) and PKCε (Figure 3C), compared to the respective non-RIPC controls. Heinen et al. [25] also did not find any changes in PKCε expression in the RIPC group of young animals under baseline conditions. However, our findings are not consistent with the study by Donato et al. [29] who showed that RIPC increased both p-Akt/Akt ratio and p-eNOS/eNOS ratio in young animals (3 months) under baseline conditions. On the other hand, we found a significant increase in the p-GSK3β/GSK3β ratio at baseline in both younger and elder RIPC groups, as compared to the non-RIPC control groups (Figure 2C). These results are partially consistent with other studies. In a study by Li et al. [45], the authors also confirmed an increase in the p-GSK3β/GSK3β ratio in the RIPC group in the young (10-week-old) mice under baseline conditions.

These studies were focused on the effect of RIPC, and not age, on the activation of RISK pathway. The only study on the efficacy of RIPC in relationship with age is the recent study by Heinen et al. [34], demonstrating that transfer of plasma from elderly patients (60–80 years) after RIPC to young rats (2–3 months) resulted in a reduction in p-GSK3β activation under baseline conditions.

It is still unclear what cardioprotective factor/factors are transferred from the remote organ to the heart by RIPC, and how maturation affects the transfer. We believe that our findings may suggest a different timing of activation of individual proteins as well as a difference in balance between recruitment of different cell survival pathways involved in cardioprotection by RIPC in relation to maturation.

### 3.2. Effect of Maturation and Remote Preconditioning on Pro- and Anti-Apoptotic Mechanisms

Studies examining the effect of age on the BAX/Bcl-2 are inconsistent. Different from our study, Liu et al. [46] found that BAX/Bcl-2 ratio did not change with increasing age (4 vs. 19 months). On the other hand, Phaneuf et al. [47] showed that under baseline conditions, Bcl-2 expression decreased with age (6, 18 and 24 months), but BAX protein levels did not change. However, our findings are in accordance with the study by Kwak et al. [48], in which the authors demonstrated not only an increase in the BAX/Bcl-2 ratio but also an increase in caspase 3 expression in the rat hearts under baseline conditions with increasing age. The decrease in the BAX/Bcl-2 ratio in the RIPC group of young rats under baseline conditions has been also confirmed, however, the experiments were performed on cell cultures [49].

The last of our main goals was to study the effect of maturation and RIPC on proteins involved in pro- and anti-apoptotic cascades in rat myocardium after I/R. With increasing age, there was a significant increase only in the ratio of BAX/Bcl-2 (in the control groups after I/R; Figure 5B), while pro-caspase 3 and caspase 3 levels did not change with age (Figure 4B,D). In RIPC-adapted animals, only age-dependent changes in pro-caspase 3 levels and its decrease after I/R were observed (Figure 4B). Caspase 3 levels and the BAX/Bcl-2 ratio did not change with increasing age in the RIPC-adapted groups of animals (Figure 4D and Figure 5B). We observed a decrease in caspase 3 expression and BAX/Bcl-2 ratio at 6 months of age in the RIPC group, compared to respective non-RIPC controls (Figure 4D and Figure 5B).

Expression of proteins involved in pro- and anti-apoptotic cascades was examined only in a few studies. However, our results are consistent with a study by Liu et al. [46], who found an increase in BAX/Bcl-2 ratio after I/R with age.

Our results are in accordance with several recent studies that have demonstrated a positive effect of RIPC on the RISK pathway proteins, as well as the proteins of the pro- and anti-apoptotic cascades after I/R in young animals. In a study by Wang et al. [49], the authors found an increase in the p-Akt/Akt ratio and levels of total PKCε in the RIPC group in young rats (after I/R). The increase in the p-Akt/Akt ratio as well as the p-GSK3β/GSK3β ratio after I/R in the RIPC group of young animals was also confirmed by other studies [31,38,39,50,51,52].

The positive effect of RIPC on the increase in the p-Akt/Akt protein ratio as well as the p-GSK3β/GSK3β ratio after I/R was also confirmed by studies in young mice [45], young pigs [53] or in young rabbits, when RIPC was applied to the aorta carotis [54]. The effect of RIPC on eNOS protein levels after I/R has not been investigated. However, in a study by Andreadou et al. [54], they showed an increase in the p-eNOS/eNOS ratio after application of RIPC to the aorta carotis in young rabbits. Previous studies examining the effect of RIPC on the ratio of BAX/Bcl-2 and caspase 3 proteins after I/R in young animals are also consistent with our results. An increase in Bcl-2 and Bcl-2/Bax ratio and a decrease in BAX and caspase 3 levels after RIPC application in young rats (after I/R) were also demonstrated in the studies by Zhang et al. [32] and You et al. [33]. A decrease in caspase 3 levels in the RIPC group of young animals after I/R was also found in a study by Ma et al. [51]. An increase in the levels of Bcl-2 protein, ratio of Bcl-2/BAX protein and a decrease in caspase 3 and Bax levels after I/R in young animals were also demonstrated by studies investigating the molecular basis of postconditioning, or liver RIPC [50,52,55,56].

Importantly, in our study, different from Akt phoshorylation, GSK3β was phosphorylated (inactivated) not only at baseline (after RIPC) but also post-I/R in both age groups, in comparison with respective non-RIPC controls. That may suggest a GSK3β-mediated effect on MPTP opening, as the final target of cardioprotection. This is also consistent with the role of mitochondria in intrinsic pro-apoptotic stimuli, and the present findings regarding the pro- and anti-apoptotic protein.

The changes in the above-mentioned proteins induced by RIPC in rats subjected to I/R in relation to age are summarized in Figure 6.

## 4. Materials and Methods

### 4.1. Animals

Three-month-old (young adults) and 6-month-old (mature adults) male Wistar rats were utilized (total number 58). Animals (250–500 g body weight) were fed a standard pellet diet and had access to drinking water ad libitum. All experiments were performed in accordance with the “Guide for the Care and Use of Laboratory Animals” published by the US National Institutes of Health (NIH) (Guide, NRC 2011) and approved by the Animal Health and Welfare Division of the State Veterinary and Food Administration of the Slovak Republic and ethics committee of the CEM SAS.

### 4.2. Perfusion Technique

Rats were anesthetized with thiopental (50–60 mg/kg) administered intraperitoneally together with heparin (500 IU). The hearts were rapidly excised, placed in ice-cold perfusion buffer, cannulated via the aorta and perfused in the Langendorff mode at a constant perfusion pressure of 73 mmHg and constant temperature 37 °C. The perfusion solution was a modified Krebs–Henseleit buffer gassed with O_2_ and CO_2_ (pH 7.4) containing (in mM): glucose 11.0; CaCl_2_ 1.6; NaCl 118.0; NaHCO_3_ 25.0; MgSO_4_ 1.18; KH_2_PO_4_ 1.28; KCl 3.0.

An epicardial electrogram (EG) was registered by means of two stainless steel electrodes attached to the apex of the heart and the aortic cannula and continuously recorded. Heart rate was calculated from the EG. Left ventricular (LV) pressure was measured by means of a non-elastic water-filled balloon inserted into the left ventricle via the left atrium and connected to a pressure transducer (MLP844, ADInstruments, Germany). LV systolic pressure (LVSP), LV diastolic pressure (LVEDP), LV developed pressure (LVDP, systolic minus diastolic pressure), maximal rates of pressure development (+(dP/dt)max) and fall (−(dP/dt)max) as the indexes of contraction and relaxation, as well as the heart rate (HR) and coronary flow (CF) were used to assess cardiac function that was analyzed using PowerLab/8SP Chart 7 software (ADInstruments, Germany).

### 4.3. Experimental Protocols

The hearts of all experimental groups (n = 8–12 hearts per group) were assigned to the following protocols (Figure 7). In the protocol of ischemia and reperfusion (A), after a 20 min stabilization period, the hearts underwent 30 min global ischemia followed by 2 h reperfusion. Remote ischemic preconditioning was performed as described earlier [16]. After anesthesia of rats, RIPC was induced by inflation (at 200 mmHg) and deflation of pressure cuff for rats (D.E. Hokanson, Inc., Bellevue, WA, USA) placed on the right hind limb. RIPC protocol (B) consisted of 3 cycles of 5 min ischemia (cuff inflation) and 5 min reperfusion (cuff deflation) (altogether 30 min). After RIPC, the hearts were excised, Langendorff-perfused and subjected to ischemia/reperfusion.

### 4.4. Induction of Ischemia/Reperfusion

Global ischemia was induced by clamping of aortic inflow for 30 min and followed by its unclamping for the evaluation of post-ischemic recovery of contractile function after 40 min reperfusion (expressed in percentage of pre-ischemic values) and determination of the size of myocardial infarction as the primary end point of injury, after 2 h reperfusion.

### 4.5. Quantification of Arrhythmias

Susceptibility to reperfusion-induced malignant ventricular tachyarrhythmias, such as ventricular tachycardia (VT) was evaluated during 10 min reperfusion [57].

### 4.6. Determination of Infarct Size

The size of the infarcted area was delineated by staining with 2,3,5-triphenyltetrazolium chloride. After staining, the hearts were cut perpendicularly to the long axis of the heart, into 1 mm thick slices, and total and unstained area were determined by a computerized planimetric method as described previously [58]. The infarct size (IS) was expressed as percentage of the area at risk (AR) size, that in the model of global ischemia represents the whole area of the left ventricle (LV).

### 4.7. Preparation of Tissue Protein Fractions

In additional parallel experiments, the tissue samples used for Western blot analysis were obtained from the left ventricle (LV) of hearts of all experimental groups (n = 4–5 per group): after stabilization (control—baseline samples) and after 40 min of post-ischemic reperfusion (control—I/R samples) or baseline and I/R samples with prior remote preconditioning (RIPC—baseline and RIPC—I/R samples). The tissues were wiped in liquid nitrogen, resuspended in ice-cold buffer A containing (in mmol/L): 20 Tris–HCl, 250 sucrose, 1.0 EGTA, 1.0 dithiothreitol (DTT), 1.0 phenylmethylsulphonylfluoride (PMSF), and 0.5 sodium orthovanadate (pH 7.4), and homogenized with a Teflon homogenizer. The homogenates were centrifuged at 800× *g* for 5 min at 4 °C. Pellets after the first centrifugation were discarded and the supernatants were centrifuged again at 16 100× *g* for 30 min. The supernatants after the second centrifugation, termed as cytosolic fraction, were used for further analysis. The protein concentrations were estimated by the method of Bradford [59].

### 4.8. Electrophoresis and Western Blot Analysis

Samples of the protein fractions containing equivalent amounts of proteins per lane (70 μg per lane) were separated by 10% or 12% SDS-PAGE gel electrophoresis (according to molecular weight of specific protein). For Western blot assays, proteins were transferred to a nitrocellulose membrane. The quality of the transfer was controlled by Ponceau S staining of nitrocellulose membranes after the transfer. Specific anti-Akt 1/2/3 (dilution 1:330), anti-p-Akt 1/2/3 (dilution 1:710), anti-eNOS (dilution 1:200), anti-PKCε (dilution 1:1000), anti-GSK3B (dilution 1:330), anti p-GSK3B (dilution 1:250), anti-BAX (dilution 1:200), anti-Bcl-2 (dilution 1:200) and anti-caspase 3 (dilution 1:100) (Santa Cruz Biotechnology) antibodies were used for the primary immunodetection. Peroxidase-labeled anti-rabbit immunoglobulin (Cell Signaling Technology) a peroxidase-labeled anti-mouse immunoglobulin were used as the secondary antibodies (dilution 1:2000). Bound antibodies were detected by the enhanced chemiluminescence (ECL) method. The optical density of individual bands was analyzed by PCBAS 2.08e software and normalized to GAPDH (anti-GAPDH antibody, dilution 1:750) as an internal control.

### 4.9. Statistical Evaluation

The data are expressed as means ± S.E.M. Two-way ANOVA and subsequent Student–Newman Keuls test, as well as Mann–Whitney U test using GraphPad Prism version 6.00 (GraphPad Software, San Diego, CA, USA) for Windows (Microsoft Corporation, Redmond, WA, USA) were used where appropriate. Differences were considered as significant at *p* < 0.05.

## 5. Conclusions

In conclusion, this is one of the few studies demonstrating that aging significantly affected the resistance of the rat heart against I/R injury during the process of maturation. We have shown not only an increase in the size of the infarction with increasing age, but also an increased susceptibility to ventricular arrhythmias. We have also found that the beneficial effect of RIPC was preserved in the elder animals, manifested as a significant infarct size limitation, improvement of LVDP recovery and decreased arrhythmogenesis.

Finally, we were able to identify signaling pathways involved in the mechanisms of cardioprotection, afforded by RIPC in relationship with aging, and to show the effect of maturation on its efficacy on the molecular levels. With increasing age, we observed a decrease in all studied proteins at baseline and after I/R: p-Akt, eNOS and PKCε, as well as p-GSK3β. After comparison of the effect of RIPC on the expression or activation of some cell survival pathway proteins, as well as of the pro- and anti-apoptotic proteins under baseline conditions and after I/R, we can conclude that some proteins involved in pro-/anti-apoptotic effects, such as BAX, Bcl-2 and caspase 3 are already activated under basal conditions, whereas “survival” proteins, such as p-Akt, eNOS and PKCε, were only significantly increased after I/R. We thus demonstrated that not only classical IPC but also RIPC showed a “biphasic” effect when survival proteins were activated in the “trigger phase” under baseline pre-ischemic conditions but also after I/R. The most important finding of this study was the finding of the increased GSK3β phosphorylation (inactivation) immediately after RIPC prior to ischemia, and also post-I/R in both age groups in comparison with non-RIPC controls. The latter may lead to the inhibition of opening of MPTP as the main source of cell death and a potential RIPC-mediated target of cardioprotection. In addition, the “biphasic” effect of RIPC on the pro- and anti-apoptotic cascades (BAX/Bcl-2) was also demonstrated.

Taken together, the results of our study support the concept that beneficial effects of RIPC may represent clinically relevant cardioprotective intervention in the elder population.

## Figures and Tables

**Figure 1 ijms-22-11009-f001:**
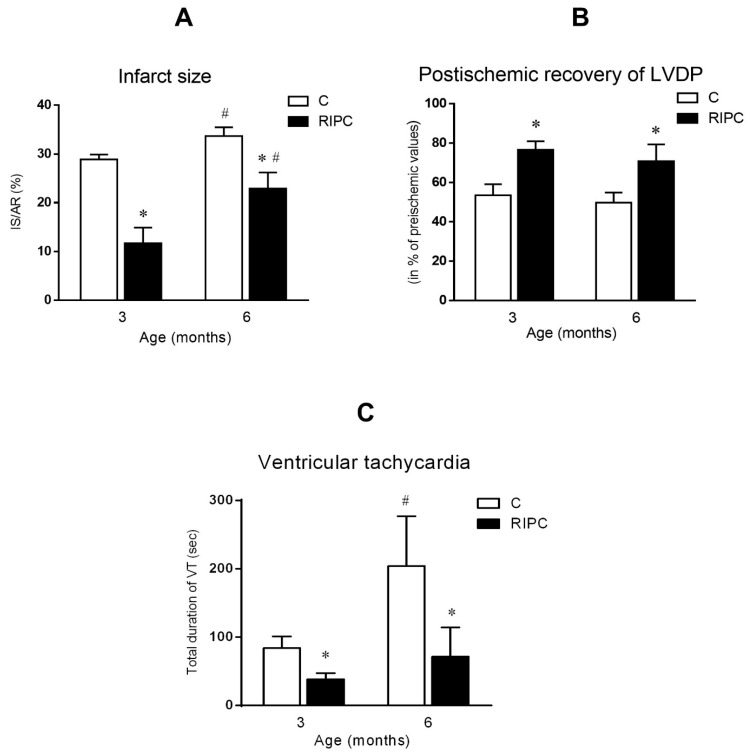
The effect of maturation on the ischemia/reperfusion injury in the non-adapted control and adapted (remote preconditioned) rat hearts. (**A**)—size of infarction. (**B**)—post-ischemic recovery of function. (**C**)—occurrence of reperfusion-induced ventricular arrhythmias. RIPC—remote ischemic preconditioning, IS—infarcted area of left ventricle, expressed in % of risk area (AR), LVDP—left ventricular developed pressure (LV systolic minus LV diastolic pressure). VT—ventricular tachycardia. Values are means ± S.E.M. from 8–12 hearts per group. * *p* < 0.05, RIPC vs. controls; # *p* < 0.05–0.01, vs. 3 months.

**Figure 2 ijms-22-11009-f002:**
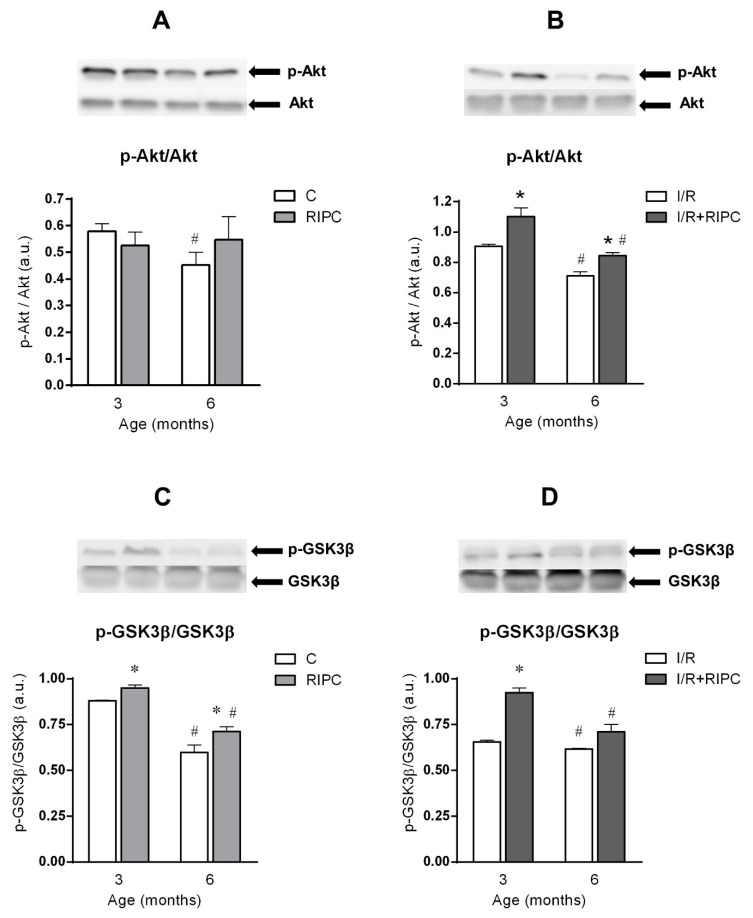
The effect of age and remote ischemic preconditioning on the expression of selected signaling pathway proteins in rat myocardium at the baseline condition (**A**,**C**) and after ischemia/reperfusion (**B**,**D**). Samples were taken prior to ischemia (after 20 min stabilization perfusion) and at 40 min of post-ischemic reperfusion. Akt—protein kinase B, p-Akt—phosphorylated (activated) Akt, GSK3β—glycogen synthase kinase 3 beta, p-GSK3β—phosphorylated (inactivated) GSK3β. The phosphorylation (activation) of Akt was expressed as the ratio of p-Akt and total Akt. The phosphorylation (inactivation) of GSK-3β was expressed as the ratio of p-GSK3β and total GSK3β. C—baseline samples, I/R—post-IR samples, RIPC—samples with prior remote ischemic preconditioning. Data are presented as means ± S.E.M of 4–5 hearts per group, normalized to the levels of GAPDH. * *p* < 0.05 RIPC vs. C; # *p* < 0.05 vs. 3 months.

**Figure 3 ijms-22-11009-f003:**
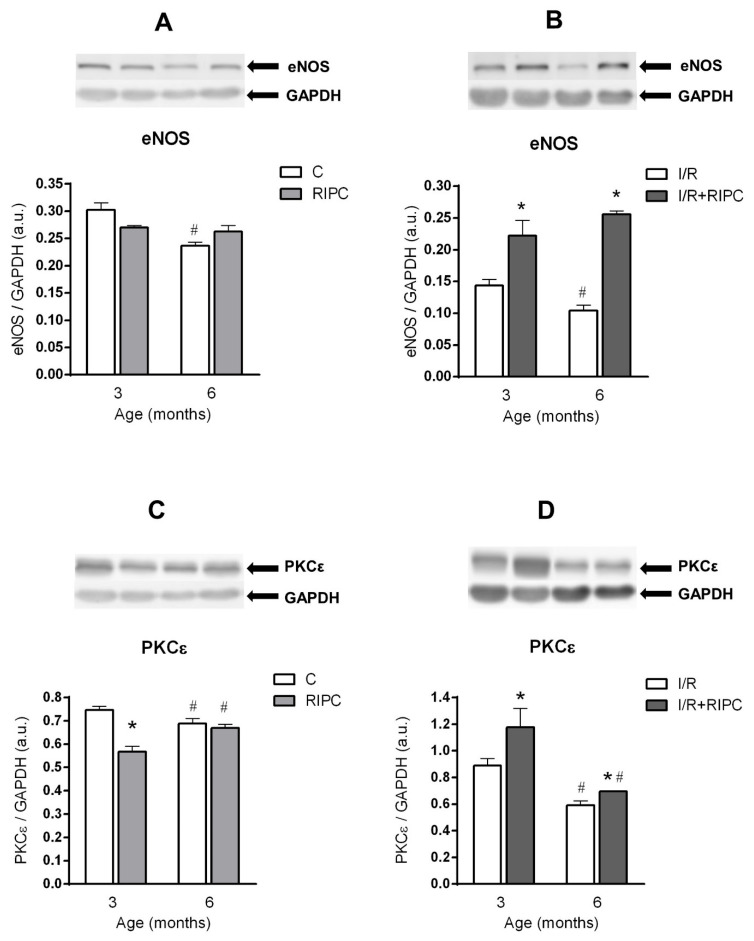
The effect of age and remote ischemic preconditioning on the expression of selected signaling pathway proteins in rat myocardium at the baseline conditions (**A**,**C**) and after ischemia/reperfusion (**B**,**D**). Samples were taken prior to ischemia (after 20 min stabilization perfusion) and at 40 min of post-ischemic reperfusion. eNOS—endothelial nitric oxide synthase, expressed in arbitrary units (a.u.). PKCε—protein kinase C epsilon, expressed in arbitrary units (a.u.). (**A**,**B**)—expression of eNOS, (**C**,**D**)—expression of PKCε. (**C**)—baseline samples, I/R—post-IR samples, RIPC—samples with prior remote ischemic preconditioning. Data are presented as means ± S.E.M of 4–5 hearts per group, normalized to the levels of GAPDH. * *p* < 0.05 RIPC vs. C; # *p* < 0.05 vs. 3 months.

**Figure 4 ijms-22-11009-f004:**
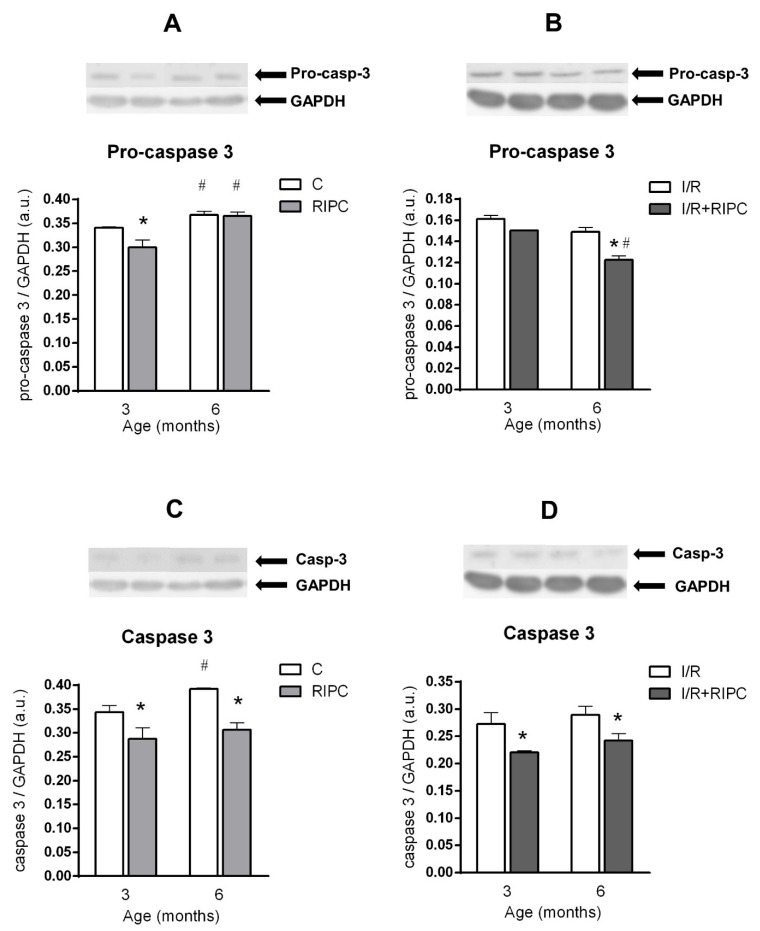
The effect of age and remote ischemic preconditioning on the expression of proteins involved in pro and anti-apoptotic cascades in rat myocardium at the baseline (pre-ischemic) conditions (**A**,**C**) and after ischemia/reperfusion (**B**,**D**). Samples were taken prior to ischemia (20 min stabilization perfusion) and at 40 min of post-ischemic reperfusion. (**A**,**B**)—pro-caspase 3, (**C**,**D**)—caspase 3. (**C**)—baseline samples, I/R—post-IR samples, RIPC—samples with prior remote ischemic preconditioning. Data are presented as means ± S.E.M of 4–5 hearts per group, normalized to the levels of GAPDH. * *p* < 0.05 RIPC vs. C; # *p* < 0.05 vs. 3 months.

**Figure 5 ijms-22-11009-f005:**
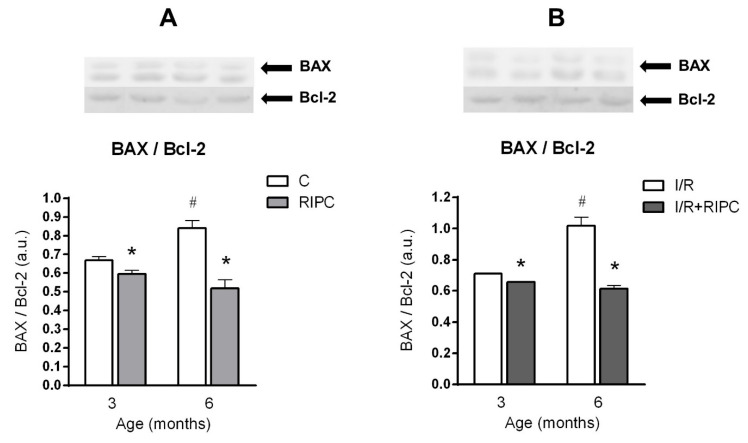
The effect of age and remote ischemic preconditioning on the expression of proteins involved in pro and anti-apoptotic cascades in rat myocardium at the baseline (pre-ischemic) conditions (**A**) and after ischemia/reperfusion (**B**). Samples were taken prior to ischemia (20 min stabilization perfusion) and at 40 min of post-ischemic reperfusion. BAX—pro-apoptotic protein from Bcl-2 protein family (B-cell lymphoma 2), Bcl-2—anti-apoptotic protein from Bcl-2 protein family. (**A**,**B**)—BAX/Bcl-2 (ratio of BAX and Bcl-2 proteins), C—baseline samples, I/R—post-IR samples, RIPC—samples with prior remote ischemic preconditioning. Data are presented as means ± S.E.M of 4–5 hearts per group, normalized to the levels of GAPDH. * *p* < 0.05 RIPC vs. C; # *p* < 0.05 vs. 3 months.

**Figure 6 ijms-22-11009-f006:**
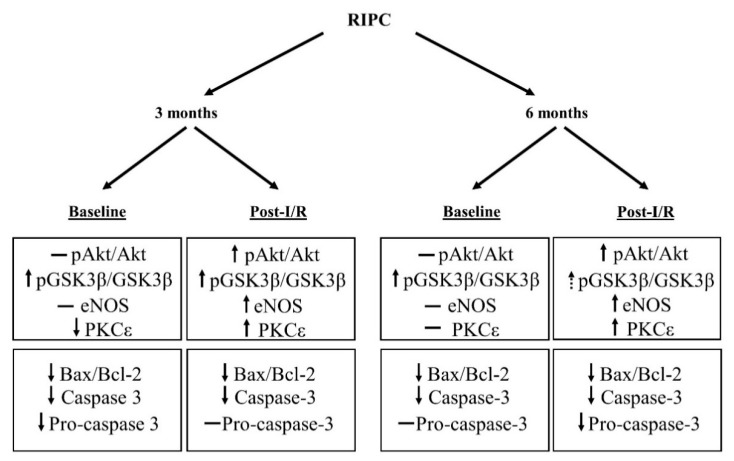
A schematic summary of the effects of RIPC on proteins of cell survival and pro/anti-apoptotic proteins under pre-ischemic baseline and post-I/R conditions in 3-month-old and 6-month-old male rats. Up arrow—significant increase, intermittent up arrow—trend to increase (*p* = 0.067), down arrow—significant decrease, straight line—no significant change.

**Figure 7 ijms-22-11009-f007:**
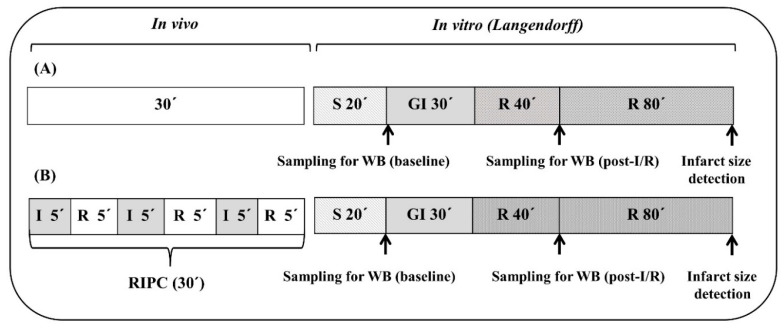
Schematic representation of the experimental protocols. (**A**)—protocol of ischemia and reperfusion, (**B**)—protocol of remote ischemic preconditioning, I—ischemia, R—reperfusion, RIPC—remote ischemic preconditioning, S—stabilization, GI—global ischemia, WB—Western blot analyses.

**Table 1 ijms-22-11009-t001:** Basic biometric parameters of rats.

Group	BW (g)	HW (mg)
3 months	250 ± 10	800 ± 14
6 months	376 ± 35 *	1100 ± 20 *

BW—body weight, HW—heart weight. Data are means ± S.E.M., n = 8–12 per group. *—*p* < 0.05, vs. 3 months.

**Table 2 ijms-22-11009-t002:** Pre-ischemic values of hemodynamic parameters of isolated rat hearts.

Group (Age)	HR (Beats/min)	LVSP (mmHg)	LVEDP (mmHg)	LVDP (mmHg)	+(dP/dt)_max_ (mmHg/s)	−(dP/dt)_max_ (mmHg/s)	CF (mL/min)
3 months	270 ± 7	87.6 ± 4.6	6.8 ± 1.7	87.0 ± 3.5	2339 ± 181	1581 ± 95	6.6 ± 0.7
6 months	261 ± 7	87.7 ± 3.8	5.9 ± 0.8	93.0 ± 3.2	2846 ± 138	1748 ± 95	8.5 ± 0.8

CF—coronary flow, LVSP—left ventricular systolic pressure, LVEDP—left ventricular end-diastolic pressure, LVDP—left ventricular developed pressure (LV systolic minus LV diastolic pressure), +(dP/dt)max, −(dP/dt)max—maximal rates of pressure development and fall, respectively, HR—heart rate. Data are means ± S.E.M., n = 8–12 per group.

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
