# Peer review of "Impact of Maturation on Myocardial Response to Ischemia and the Effectiveness of Remote Preconditioning in Male Rats"

_ijms, 2021, doi:10.3390/ijms222011009_

Round 1
Reviewer 1 Report
In this study, the authors evaluated the age-dependent mechanisms of the protective effects of remote ischemic preconditioning (RIPC) in hearts from young and mature rats in Langerdorff systems. The authors demonstrated that mature aging did not affect cardiac function at baseline but it affected response to ischemia and reperfusion injury. At molecular analysis, cardiac dysfunction was associated with alteration of several RISK proteins in an age-dependent manner. The manuscript contains interesting data about the mechanism of RIPC. Data also support the conclusion. Some issues, however, should be addressed.
Major comments:
- Abstract: Several sentences (lines 22-29, page 1) are not clear and should be re-phrased.
- Methods and Results: The authors evaluated kinase and eNOS expression in cytosol fraction only. Is there a reason for excluding the nuclear compartment? Many of these kinases undergo to a nuclear-cytosol shuttling, therefore, their decrease in the cytosol may represent an increase in the nucleus. The authors should also provide nuclear expression of the RISK kinases.
- Figures 2 and 3 and Figures 4 and 5: For a better appreciation of the RISK and apoptosis changes, the baseline data and the post-injury data for each kinase should be shown in the same panel.
- Figure 6. The size of bars is arbitrary and misleading. It would be helpful if the authors provide a scale on the x axis. Also, it is not clear whether the 6-month controls are normalized or compared to young control.
- Figure 7. The time point for western blot analysis is not clear. Were kinases and apoptotic proteins determined at both 20 and 40 minutes after RIPC? What was the time point for baseline measurements? These time points should also be defined in the figure legends of Figures 2, 3, 4 and 5.
Reviewer 2 Report
Dear Authors,
The manuscript reports the impact of the age on the myocardial response to ischemia and heart preconditioning in rats. The age of the patients but also the pretreatment of the organs play a major role in the organ protection against the ischemia-reperfusion injuries.
I have comments:
- Line 139 as example to be checked all over the manuscript: “decrease in the expression of phosphorylated (activated) Akt”. This sentence is incorrect. In fact the expression of Akt in this case is decreased over aged, but the level of phosphorylation (which doesn’t change the level of protein expression ) is unchanged in the 3 months age group, but the total AKT expression is decreased between 3 and 6 months.
- The Western Blot data on figure 2, 3, 4 and 5 are not clear. IN the figure 7, 4 different samples are collected for Western blots, at 2 different time and in 2 different groups, in 2 different conditions. It means that 8 samples should be compared per figure when only 4 samples are analyzed, excluding the basal expression of the proteins. All these figures legend are reporting the proteins level changes related to the age but also to the ischemic preconditioning. It is very difficult to understand what the authors are presenting, Based on this, It is very difficult for me to understand and analyze the data.
- Figure 6: The colors should be explained in the legend to understand the color meaning and the length meaning. Also, as mentioned in my previous comment for the western blot, this table confirms that 8 samples should be compared in the western blot and it’s not the case. Data are missing.
- The IACUC protocol number must be provided
- How many rats were used and per group?
- Number of the chemical compounds should be subscript. (CaCl2 should be CaCl2) This should be corrected all over the manuscript.
Sincerely
Round 2
Reviewer 1 Report
No further comments
Reviewer 2 Report
Dear Authors,
I have no additional comments.
Sincerely